# Selective auditory attention modulates cortical responses to sound location change for speech in quiet and in babble

Erol J. Ozmeral[1]*, Katherine N. Menon[2]

1 Department of Communication Sciences and Disorders, University of South Florida, Tampa, FL, United States of America, 2 Department of Hearing and Speech Sciences, University of Maryland, College Park, MD, United States of America

* eozmeral@usf.edu

**Data Availability Statement:** All relevant data are within the paper and its Supporting Information files.

**Funding:** This work was supported by an internal grant from the College of Behavioral and

## Abstract

Listeners use the spatial location or change in spatial location of coherent acoustic cues to aid in auditory object formation. From stimulus-evoked onset responses in normal-hearing listeners using electroencephalography (EEG), we have previously shown measurable tuning to stimuli changing location in quiet, revealing a potential window into the cortical representations of auditory scene analysis. These earlier studies used non-fluctuating, spectrally narrow stimuli, so it was still unknown whether previous observations would translate to speech stimuli, and whether responses would be preserved for stimuli in the presence of background maskers. To examine the effects that selective auditory attention and interferers have on object formation, we measured cortical responses to speech changing location in the free field with and without background babble (+6 dB SNR) during both passive and active conditions. Active conditions required listeners to respond to the onset of the speech stream when it occurred at a new location, explicitly indicating 'yes' or 'no' to whether the stimulus occurred at a block-specific location either 30 degrees to the left or right of midline. In the aggregate, results show similar evoked responses to speech stimuli changing location in quiet compared to babble background. However, the effect of the two background environments diverges somewhat when considering the magnitude and direction of the location change and where the subject was attending. In quiet, attention to the right hemifield appeared to evoke a stronger response than attention to the left hemifield when speech shifted in the rightward direction. No such difference was found in babble conditions. Therefore, consistent with challenges associated with cocktail party listening, directed spatial attention could be compromised in the presence of stimulus noise and likely leads to poorer use of spatial cues in auditory streaming.

## Introduction

Our ability to perceive auditory motion has been studied for over a century [1], and its relevance to virtual and augmented auditory spaces and hearing device technologies has led to a

Community Sciences at the University of South Florida and by the National Institute of Deafness and Other Communication Disorders (NIDCD; https://www.nidcd.nih.gov/) to E.J.O. (R21 DC017832). The funders had no role in study design, data collection and analysis, decision to publish, or preparation of the manuscript.

**Competing interests:** The authors have declared that no competing interests exist.

resurgence in interest in the last couple of decades. Psychophysically, listeners are able to detect a change in the spatial location of a stimulus with as little as 1˚-resolution at the front azimuth but perform worse for stimuli that change location off-of-center (between 1.5˚ and 11.3˚ when the reference stimulus is at 60˚ and depending on the center frequency and bandwidth; i.e., minimum audible angle [MAA]) [2, 3]. For stimuli that move at a fixed velocity, the smallest average arc to detect the movement (i.e., minimum audible movement angle [MAMA]) [3, 4] tends to be greater than the MAA with poorer performance associated with faster velocities. Neural imaging has also shown that responses to location changes are dependent on the reference location and extent of a shift in the free field [5–7] or lateral position in the case of binaural stimuli over headphones [8–11]. Understanding the neural mechanisms associated with spatial hearing, auditory motion perception, and selective attention offers unique opportunities to address fundamental challenges that listeners with hearing loss, mainly older, face daily.

Natural acoustic environments often include dynamic sources in their relative location to the listener. In the classic "cocktail party" [12] and other substantiations [13, 14], most normal-hearing listeners are adept at following speech and other auditory objects to maximize communication goals and awareness of potential threats. Much like visual attention, auditory attention relies on the ability to perceptually group objects of which we can then decide to emphasize or ignore [15, 16]. Many complex, sequential cognitive processes are required to achieve this object-based auditory attention. The brain must first be able to form distinct auditory objects based on common spectro-temporal properties of stimuli before segregating between foreground and background [17]. Over time, this streaming process is subject to top-down modulatory effects of attention, which manifests as increased neural representation for sensory stimuli features (referred to as sensory gain control) [18, 19]. To successfully follow speech signals amid competing background stimuli, therefore, it is thought that the brain can attend to relevant signal channels both by adding gain to encoding of relevant signals and "filtering out" the extraneous signals [20, 21].

In an earlier study, we investigated the effects of selective attention on an auditory event-related potential (AERP) [22] evoked by narrowband noise stimuli that changed locations in the front horizontal plane [7]. Active attention to a distinct spatial location was predicted to yield sharper tuning to spatial changes at or near the target location. The results of this previous study were consistent with sensory gain control; however, the stimulus construct was far from what would be considered a "cocktail party," and therefore, it was unknown whether similar effects could be observed with more ecologically relevant stimuli. It may be that such neural responses would be quite different for speech stimuli, and the addition of energetic noise could potentially eliminate any evoked responses to spatial change. On the other hand, if similar responses are indeed observable with and without background interferers, there is the potential to use this objective measure as a conduit for measuring successful auditory object formation in complex acoustic scenes.

The approach of the present study combines behavioral and electrophysiological measures to investigate the dynamic modulation of sensory-evoked brain activity by spatial attention. If objects are successfully formed, attention to a perceptual feature like location is presumed to modulate the neural activity related to changes in that attended perceptual feature. To that end, the difference in effects between an arbitrary signal (like noise), versus a speech stream was investigated while evaluating the ways in which a favorable signal-to-noise ratio (SNR) affects neural responses to speech changing locations. The main hypothesis was that attention to the location of a speech stream would evoke stronger responses to spatial changes than passive listening, much as was observed with the less complex stimuli in previous studies. Due to the inherent challenges evident when acoustic scenes are complex, an additional goal was to

test the hypothesis that the addition of background babble would reduce the observed effects of spatial attention in quiet.

## Materials and methods

### Participants

Participants included 18 adults (16 females) between the ages of 18 and 25 years of age (mean: 21.6) with audiometrically normal hearing ($\leq$ 20 dB HL at octave frequencies between 250 and 8000 Hz). Data collection was completed over the course of 3–4 visits lasting approximately 2 hours each in duration. Exclusion from participating included any reported history of neurological dysfunction, middle ear disease, or ear surgery. The Montreal Cognitive Assessment (MoCA) was administered to all participants to screen for cognitive impairment, and all listeners had a passing score of at least 26 [23]. All participants provided written consent for study participation prior to testing, and all procedures were approved by the University of South Florida Institutional Review Board. Participants were compensated for their time at an hourly rate.

### Stimulus presentation

Target stimuli consisted of monosyllabic English words (2535 tokens) recorded from a male speaker with 100-ms inter-stimulus intervals at 76 dB SPL. In Experiment I, stimuli were presented in quiet. In Experiment II, stimuli were presented in multi-talker babble, consisting of eight turn-taking conversations spoken in eight foreign languages at 70 dB SPL overall (+6 dB SNR; for more details on target and background stimuli, see [24]). A foreign language multi-talker babble was chosen for its more natural properties relative to steady-state noise, but it's important to note that any masking by the stimulus would be primarily energetic [25]. Stimulus files (.wav) were loaded in MATLAB (MathWorks, Natick, MA) at a 44.1 kHz sampling rate and presented in the free field. Digital-to-analog conversion was performed by a 24ao (MOTU, Cambridge, MA) soundcard routed to three ne8250 amplifiers (Ashly Audio, Webster, NY) to 24 possible Q100 loudspeakers (KEF, Maidstone, England) in the azimuthal plane (Fig 1A). Target stimuli could be presented from only one location at a time, at either ±60˚, ±30˚, or 0˚ azimuth (checkered boxes in Fig 1A) and switched to a random location with replacement every 2 seconds. Catch trials were included at equal probability in which no change in location occurred. In Experiment II, background babble was presented simultaneously from locations at ±165˚, ±105˚, ±75˚, and ±15˚ azimuth (black boxes in Fig 1A). Fig 1B shows a 10-second example stimulus presentation for Experiment II with time on the x-axis and stimulus waveforms centered at their respective speaker location angle on the y-axis.

### Procedure

Testing took place in a double-walled, sound-treated booth. Listeners sat in a height-adjustable chair facing the arc of loudspeakers approximately 1 m away from the listener at ear level (Fig 1A). Throughout testing, listeners were instructed to remain as still as possible and to maintain their head position such that their nose was pointed towards the center speaker (0˚), with monitoring conducted by the experimenter through the sound booth window. To measure the effects of spatial attention, listeners were either instructed to actively attend to one of two possible speaker locations, or they were told to watch a silent video on a display in front of them during the audio presentation. In the active (Attend Left or Attend Right) conditions, the corresponding speakers were marked visually by either a blue 'x' (Attend Left) or a red 'o' (Attend Right). A touchscreen display was positioned near the participant's right hand with a user interface (all participants were right-handed). The user interface was generated in MATLAB

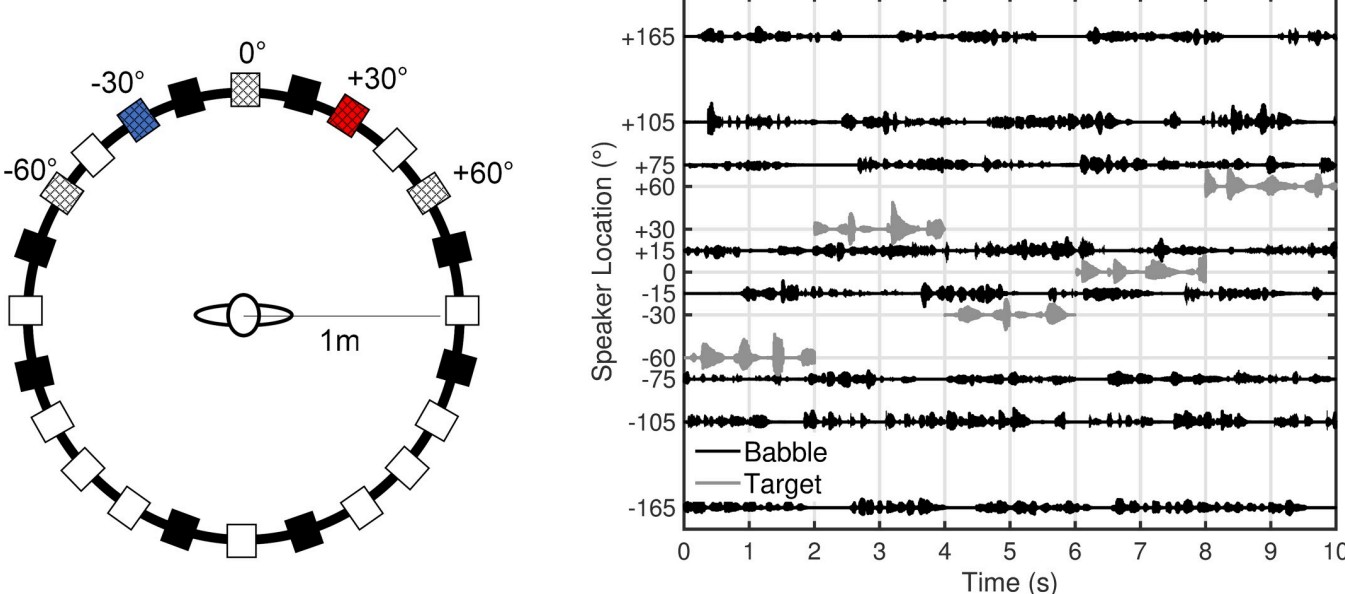

**Fig 1.** Panel A shows a schematic of the laboratory spatial array. The target speech stream came from alternate locations labeled, -60, -30, 0, +30, and +60˚. For the directed-attention conditions, listeners were instructed to either attend to the speaker at -30˚ (blue; Attend-30˚) or attend to the speaker at +30˚ (red; Attend+30˚). In Experiment I, the speech stream was presented in quiet. In Experiment II, the speech stream was presented in background speech babble at +6-dB SNR from 8 speaker locations (black boxes). Loudspeaker positions indicated by white boxes were not used in this study. Panel B shows a 10-second example presentation (Experiment II). Target (grey; 76 dB SPL) and babble (black; 70 dB SPL overall) waveforms are positioned at their respective speaker locations over time. The location of the target could change every 2 seconds. In Experiment I, only speech targets were presented.

and consisted of two buttons reading "Yes" and "No" in a horizontal row on the southeast location of the display. During active attention conditions, the participants were told to respond every time that the target stimulus changed positions with either a tap of a "Yes" button when the target moved to the location that they were attending to (16% of trials) or "No" when the target moved to any other location (64% of trials). Only trials in which the signal changed locations required a response from the participant. Catch trials (20% of trials)–those in which the stimulus location had no change–were not meant to elicit a button press for target locations. Except for the attention component (including the button press), active and passive conditions were identical in stimulus presentation and environment. Blocks of listening, with self-paced breaks in between, were 150 2-second trials.

## Electroencephalography (EEG)

Continuous EEG was recorded using a Waveguard™ (ANT Neuro BV, Enschede, Netherlands) elastic cap with 64 sintered AG/AgCl electrodes (International 10–20 electrode system). All electrode impedances were below 10 kΩ, with digitization at 512 Hz and 24-bit precision. Ground was located at the central forehead (AFz). All electrodes were initially referenced to the Cpz electrode. Recordings were made through asalab™ acquisition software (ANT Neuro, Enschede, Netherlands), and triggering was controlled using custom MATLAB scripts via the digital input/output stage of an RZ6 (System III, Tucker-Davis Technologies, Alachua, FL).

Recordings were processed offline using the software suite Brainstorm [26], running within MATLAB. Pre-processing of raw recordings consisted of band-pass filtering between 0.1 and 100 Hz (slope of 48 dB/octave), notch filtering at 60 Hz and harmonics, automatic detection of eye blinks based on the frontal electrodes [26], re-referencing to the average, and artifact removal based on spatial-source projections (SSP) [27, 28]. The SSP approach is very similar to an

independent component analysis that performs a spatial decomposition of the signals to identify the topographies of an idiosyncratic event, such as an eye blink. Because these events are reproducible and occur at the same location, this analysis can use their spatial topographies to remove their contribution from the recording while preserving contributions from other generators.

Following pre-processing stages, each continuous recording was epoched by trigger type, corresponding to shift pairs (i.e., the combination of where the stimulus occurred and where it was immediately before the location change). Trigger labels were applied at the time of measurement in which both the pre- and post-shift location were coded in the label. There were five possible target locations, so 25 possible pre- and post-shift combinations, including the catch trials. Epochs were further separated for attention conditions based on whether they were followed by either a "Yes" or "No" button press within 2 seconds of the stimulus location change. All epochs of a given pair combination (and button response) were then averaged following the removal of DC offset and linear drift. To capture the overall activity across the 64 scalp electrodes, the RMS activity, or global field power (GFP; [29]), was computed for subsequent analyses.

## Analyses and statistical approach

The N1, P2, and P3 of the AERP were identified by their latencies relative to the onset of the spatial change of the speech stimulus. To extract individual peaks, the average peaks were first measured manually for each condition, and then a peak-finding tool in MATLAB was used to extract peaks within a 40-ms window around the average latency of each of the AERP components. Once individual data had been fully processed and peaks extracted, grand averages and standard deviations were computed across all subject data for each group.

To test the hypothesis that attention to spatial location would modulate the evoked response to a spatial change, initial analyses centered on the AERPs associated with each of the five spatial locations that the speech was presented for the attention conditions. Only trials with accurate responses were considered for Attend Left and Attend Right conditions. Correct trials were determined by the button press indicating whether the target moved to an attended location or not (all trials were "correct" for the Passive condition). Accuracy was above 95% for all participants. Across each time point for the grand averages, a t-test was then performed to determine statistical differences between attention conditions at each spatial location. The results were intended to indicate at which time points attention had an effect and included each of the specific latencies mentioned above.

Following those initial analyses, it was important to determine how the magnitude of the spatial change and the direction of the spatial change affected our results. Previous studies have indicated that these metrics interact with the role of attention [30]. The late-latency P3 responses were analyzed with respect to both the pre- and post-switch location of the speech stream to test the question of whether spatial attention would modulate activity dependent on the magnitude and direction of the change in speech location. Data were collapsed across equal shift sizes and direction, independent of post-switch location, and submitted to a repeated-measures ANOVA with three levels of attention condition (Passive, Attend Left, and Attend Right) and 9 levels of shifts (-120° to 120° in 30° increments, where negative shifts were to the left and positive shifts to the right). Posthoc comparisons were Bonferroni corrected. All statistical analyses were performed in SPSS (version 28, IBM Corp.).

## Results

### Cortical response to speech stream changing location in the free field

**Experiment I: Quiet background.** Each panel in Fig 2 shows the average scalp response for the 2-second period following a change in the speech stream location in Experiment I for

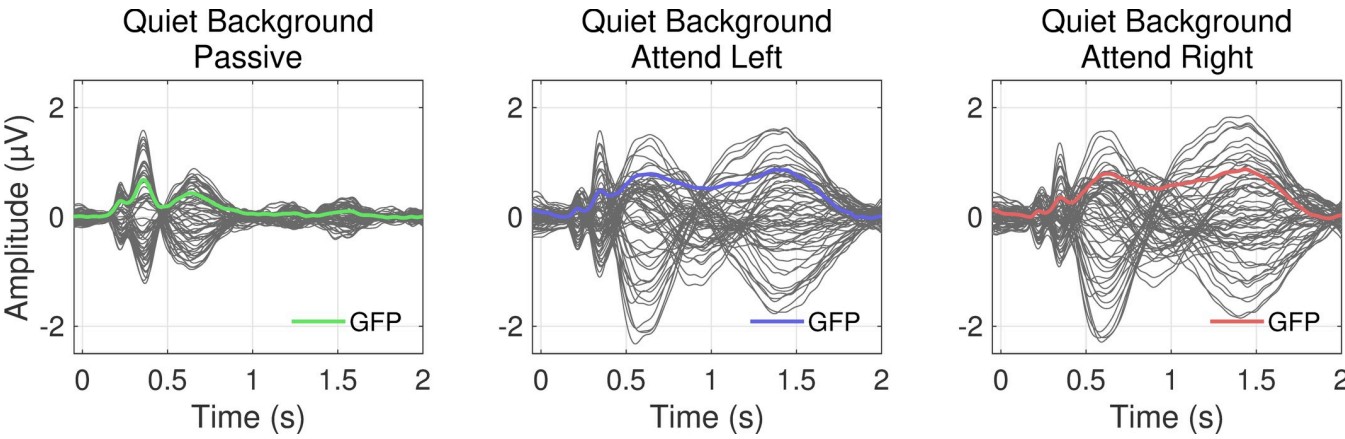

**Fig 2. Average potentials measured at the scalp for three attention conditions (Passive, Attend Left, and Attend Right) in quiet background.** Dark grey lines represent 64 recording sites, and colored lines are the baseline-corrected global field power (GFP) across all electrodes.

the three attention conditions (Passive, Attend Left, and Attend Right). Colored lines in Fig 2 panels represent the baseline-corrected global field power (GFP). Deflections near latencies of 125 ms, 250 ms, 550 ms, and around 1.4 s characterize the morphology in each panel. The two early deflections likely correspond to N1 and P2 activity in response to the change in location of the target stimulus [31, 32]. In contrast, the third peak could reflect a P3 response indicating a higher-level of awareness and uncertainty in the environmental change [33–35]. The only substantive difference between attention conditions was whether listeners responded to target location changes via button press. This likely explains the greater amplitudes at and sustained activity between the latter two latencies in the Attend Left and Attend Right versus Passive conditions.

Colored lines in Fig 3 indicate the attention condition (Passive, black; Attend Left, blue; Attend Right, red). As in Fig 2, we see morphological similarities and differences among the Passive, Attend Left, and Attend Right conditions at each azimuth. The differences in the two late deflections and overall sustained activity between the Attend Left and Attend Right conditions are most notable. At locations in which the listener responded 'no' for all correct trials in both conditions (-60°, 0°, and 60°), the average responses are aligned, indicated by seldom significant differences in a paired-sample t-test (alpha = 0.05; black horizontal markers). However, for the two possible target locations (-30° and 30°), there is a clear increase in activity in

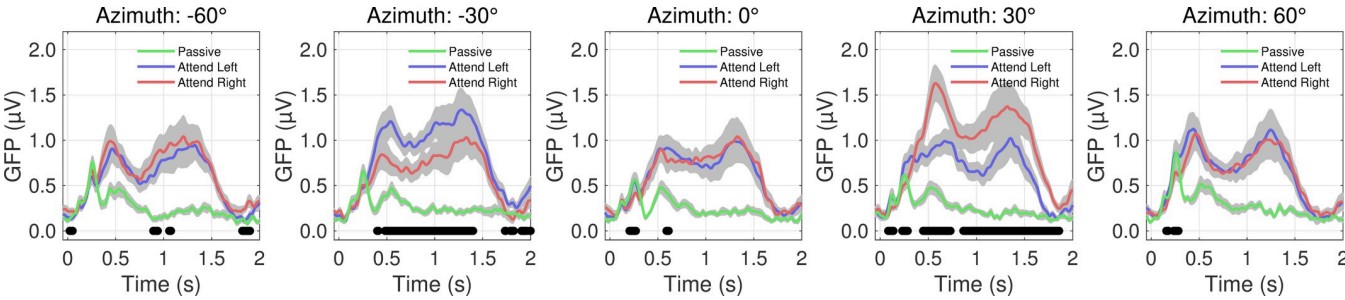

**Fig 3. Grand average global field power (GFP) for stimuli presented in quiet background for each attention condition (Passive [green]; Attend Left [blue]; Attend Right [red]).** The five panels represent trials in which the target arrived at each of the five horizontal speaker locations (from left to right: -60° to +60°). Grey shaded regions represent standard error of the mean, and black horizontal markers represent time latencies in which a significant difference was found between Attend Left and Attend Right conditions in a paired-sample t-test (alpha = 0.5; n = 18).

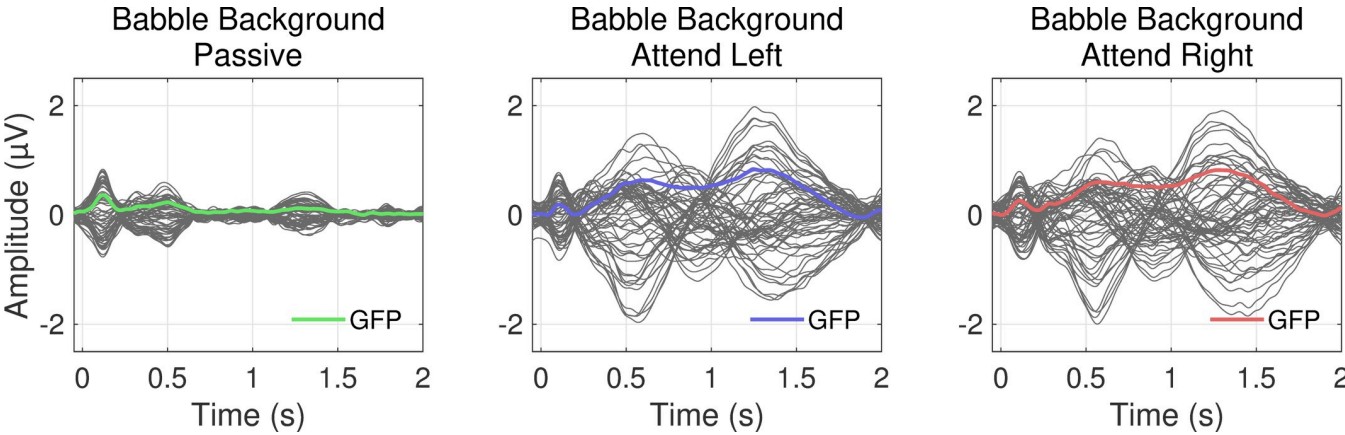

**Fig 4. Average potentials measured at the scalp for three attention conditions (Passive, Attend Left, and Attend Right) in babble background.** Dark grey lines represent 64 recording sites, and colored lines are the baseline-corrected global field power (GFP) across all electrodes.

which the listener responded 'yes' for correct trials (compare larger blue than red curve at -30˚ and larger red than blue curve at 30˚), indicated by consistent significant differences starting around 500 ms. Black horizontal markers indicate the time points that yielded significant differences in amplitude between the Attend Left and Attend Right conditions in a paired-samples t-test (alpha = 0.05).

**Experiment II: Babble background.** The addition of background babble in Experiment II posed the threat of reducing or eliminating observable cortical activity to the target stream. In Fig 4, the average responses for the 2-second period after a target location change are plotted. The morphologies of these responses to a target stream in +6 dB-SNR babble are like those in quiet (Experiment I; Fig 2), with comparable deflections at equivalent latencies. However, there were slight differences worth noting; specifically, the N1 deflection was less evident, and the responses overall were smaller compared to those in Experiment I. Fig 5 shows the activity when separated by spatial location (like Fig 3 for Experiment I). At all locations, there was a robust component around 550 ms and 1.4 s with overall greater potentials for active rather than passive attention conditions. For conditions in which listeners attended to a specific target location (i.e., ± 30˚), responses were demonstrably larger in the corresponding attention condition, whereas at non-target locations, the two attention conditions did not elicit remarkably different responses from one another.

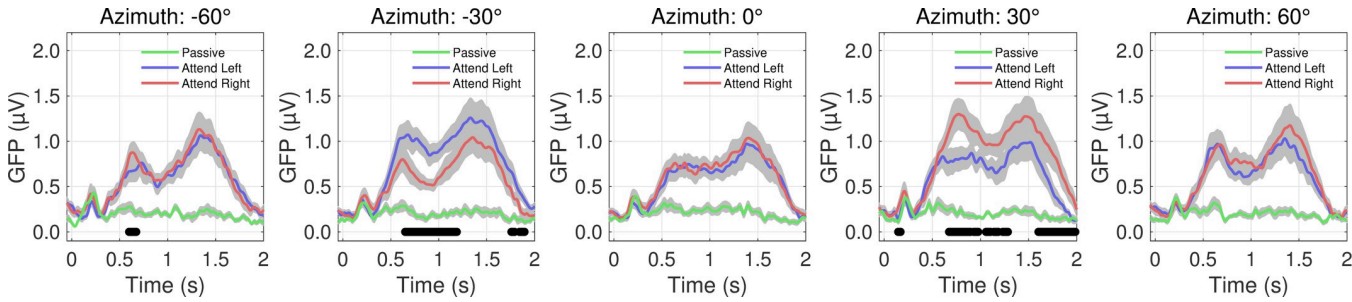

**Fig 5. Grand average global field power (GFP) for stimuli presented in babble background for each attention condition (Passive [green]; Attend Left [blue]; Attend Right [red]).** The five panels represent trials in which the target arrived at each of the five horizontal speaker locations (from left to right: -60˚ to +60˚). Grey shaded regions represent standard error of the mean, and black horizontal lines represent time latencies in which a significant difference was found between Attend Left and Attend Right conditions in a paired-samples t-test (alpha = 0.5; n = 18).

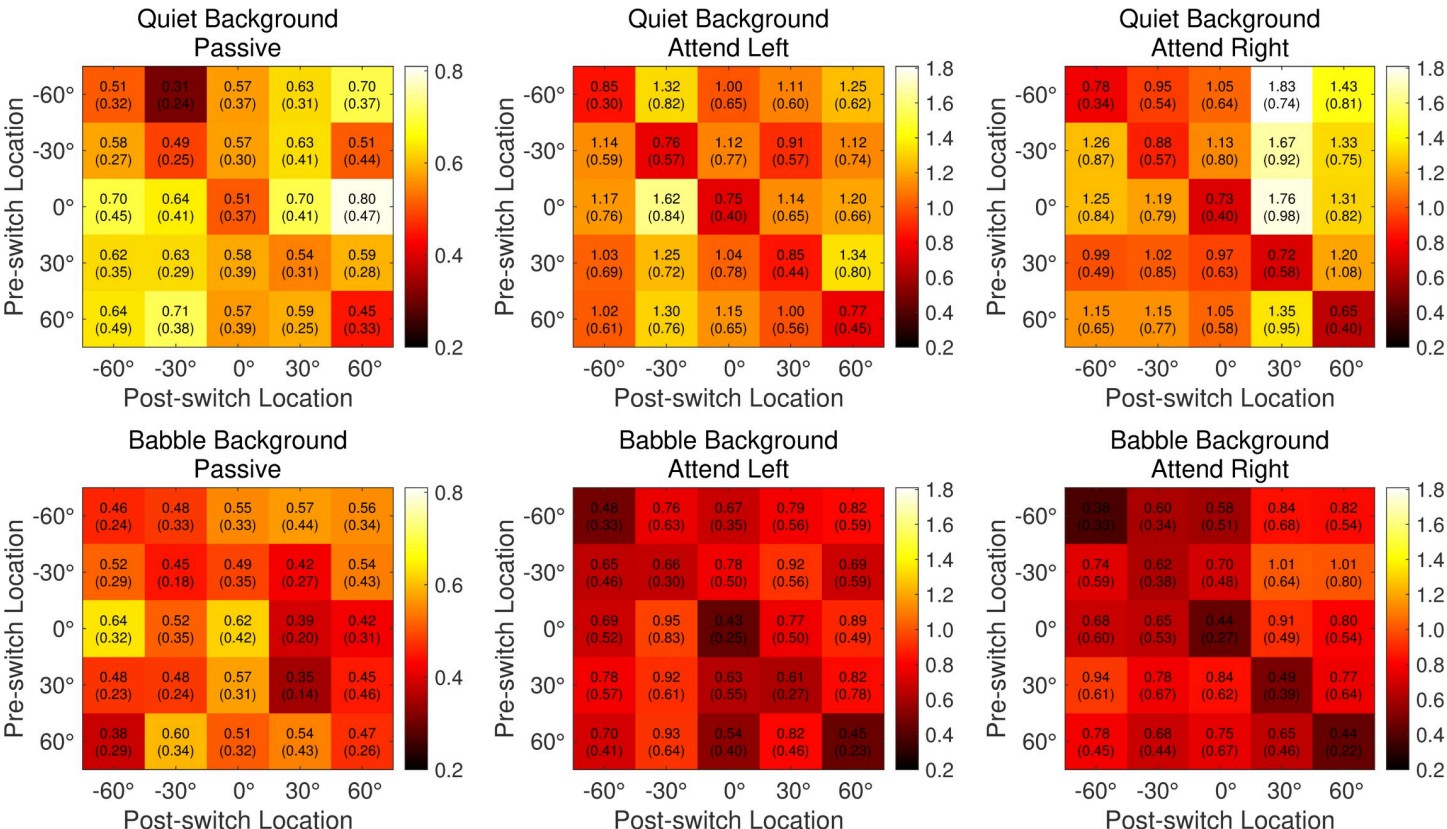

**Fig 6. Results per attention and background condition are reported as matrices of GFP activity at around 550 ms with rows indicating the pre-switch location and columns indicating the post-switch location.** Matrix cells along the diagonal from top left to bottom right represent catch trials in which stimuli had no location change. Panels on the left are for Passive conditions and have a different range of values compared to the two attention conditions (middle: Attend Left; right: Attend Right) as indicated by the color bars. The top panels are for quiet conditions and the bottom panels are for the babble conditions.

### Late-latency P3 modulated by spatial attention

In Fig 6, results per attention and background condition are reported as matrices of GFP activity at around 550 ms with rows indicating the pre-switch location and columns indicating the post-switch location. Matrix cells along the diagonal from top left to bottom right represent catch trials in which stimuli had no location change. Panels on the left are for Passive conditions and have a different range of values than the two attention conditions (middle: Attend Left; right: Attend Right), as indicated by the color bars. The top panels are for quiet conditions, and the bottom panels are for the babble conditions. Overall, as was seen in earlier analyses, there was less activity in the Passive condition than the two active attention conditions. There were generally larger responses to speech moving in quiet than in babble background. In both the quiet and babble backgrounds, for Attend Left and Attend Right, there was a tendency for larger responses for post-switch locations matching the target location at -30˚ and +30˚ respectively.

To measure the effect of magnitude and direction of the location changes shown in Fig 6, all location-change events measured around 550 ms were collected and binned according to the absolute displacement (size and direction) of the change, without regard to the pre or post switch position. The average GFP amplitude was plotted against this "spatial change" in Fig 7. It should be noted, however, that the analysis includes both target and non-target trials so it is not an independent assessment of the effects of stimulus location change. The resulting mean

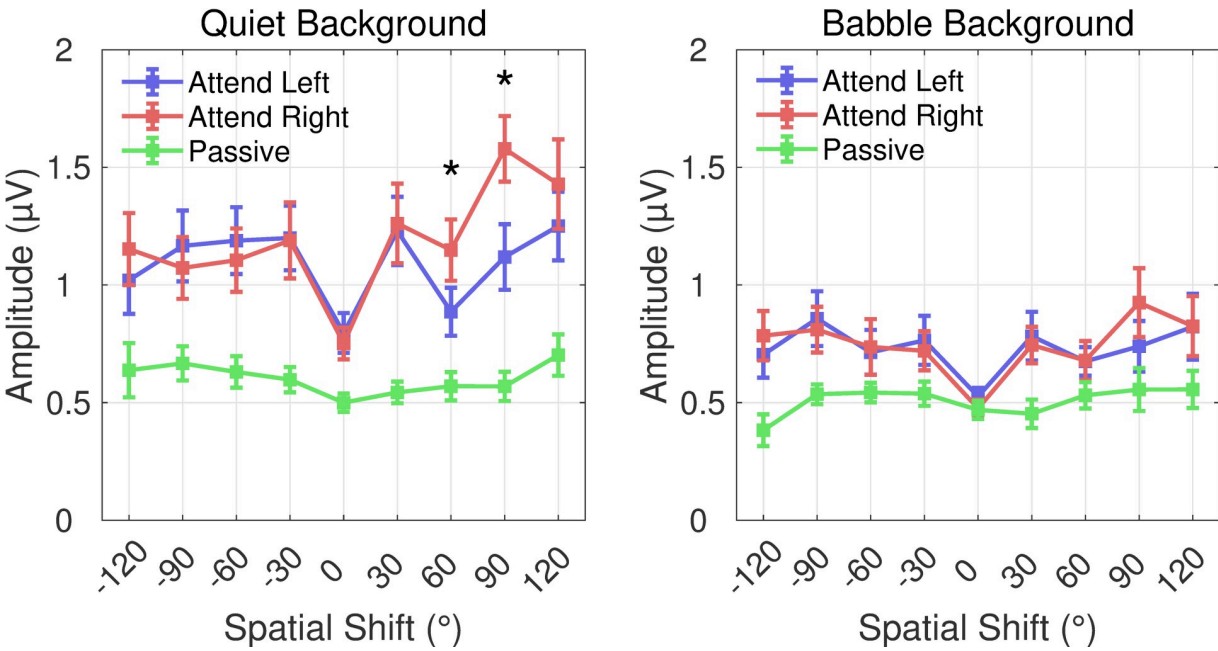

**Fig 7. The average GFP activity around 550 ms for a spatial change (displacement between pre and post switch) spanning ±120° in 30° increments.** All trials were included for the Attend Left (blue) and Attend Right (red) conditions at all locations without regard to button press. Responses in Passive trials (green) were significantly lower than the attention conditions when in Quiet (left panel) and when in Babble background (right panel).

spatial-change tuning curve is v-shaped with a minimum response for the average of the no-change catch trials and maximum responses typically near ±120° spatial shift. In separate two-way repeated-measures ANOVAs with nine levels of spatial shift and three levels of attention, there was a main effect of attention condition in each background (quiet: $F[2,34] = 24.2$, $p < .001$, $\eta_p^2 = .59$; babble: $F[2,34] = 9.1$, $p < .001$, $\eta_p^2 = .35$) and angle (quiet: $F[8,136] = 8.3$, $p < .001$, $\eta_p^2 = .33$; babble: $F[8,136] = 3.1$, $p = .003$, $\eta_p^2 = .15$). Bonferroni-corrected *posthoc* tests confirm that the amplitudes in the attention conditions were greater that the Passive condition (all $p < .01$). In addition, Bonferroni-corrected *posthoc* analyses indicated there was a significant difference between Attend Left and Attend Right for two spatial-change vectors in the rightward direction in the quiet background (indicated by asterisks in Fig 7) but no differences between the two for speech in babble.

## Discussion

In a previous study, we explored the neural modulatory effects of spatial attention on a narrowband noise burst periodically changing between five locations in the front horizontal plane [7]. We observed a clear effect of attending to one spatial hemifield versus the other, such that location-change responses were largest at the attended location, and all active attention conditions yielded stronger cortical responses than passive conditions overall in both younger and older normal-hearing adults. The present study similarly tested the effects of spatial attention on evoked scalp responses to a speech stream in younger, normal-hearing listeners because it was unknown whether more spectro-temporally complex stimuli with greater ecological relevance would elicit comparable location-change responses. Also, of interest was whether a babble background would disrupt modulatory effects of attention on the speech stream, which has the potential to explain mechanisms associated with cocktail party listening.

The morphology of the electroencephalogram for speech changing location in quiet and babble background included sensory potentials, N1 and P2, that were not modulated by attention. The N1 component was less prominent when background babble was present, which is consistent with previous research on the effects of signal-to-noise ratio on cortical responses to speech in noise [36]. In all conditions, the N1 and P2 components were slightly delayed relative to classic sensory potentials to the onset of stimulus energy [37, 38]. Previous studies that have measured EEG responses to location changes have also shown delayed latencies relative to those elicited by stimulus onsets and have labeled these components as "change"-N1 or "change"-P2 that make up the motion-onset-response (MOR) [39, 40]. Though the MOR is influenced by stimulus-dependent factors [41–43], it is also thought to reflect higher-level auditory areas and can change with task-relevant attentional processes [44]. We did not observe significant differences between the attention and passive conditions in the present study at these earlier latencies. Still, there are key differences between this study and those that specifically explore auditory motion. The first major methodological difference is that the present stimulus construct included an instantaneous randomized location change rather than a successive change in location to each loudspeaker along a motion path. This "jump" in location may have precluded any attentional effects on auditory motion. A second key difference is that our study required listeners to actively respond to changes in the stimulus location only in the attention conditions and therefore lacked a motor control in the Passive conditions. Previous studies included a non-spatial or irrelevant task in the baseline condition (e.g., [44]), and therefore, any differences at N1 or P2 could be attributed to differences in attentional processes. Thus, the present study focused more on the varied responses between the Attend Left and Attend Right conditions, which differed by the location listeners were required to attend.

For the passive and both attention conditions, P3 amplitudes were small for catch trials, in which no location change occurred 20% of the time, and they were large for location changes. This was consistent with our earlier study using narrowband noise stimuli [7]. The fact that a late-latency deflection was observed in the Passive condition also suggests that despite instruction to ignore the speech during these blocks, the discrete changes in location of the speech stream were salient, and this response may reflect an exogenous capture of attention [45, 46]. Early studies of the P3 often describe it as being elicited by surprising or unexpected stimuli or stimulus changes (e.g., [47]), and it has been shown to be larger for task-relevant stimuli requiring a response rather than unattended or irrelevant stimuli [48, 49]. Here, the late-latency P3 response is believed to be associated with higher-order processing related to its uncertain change in location (80% of the trials had equal probability to change to one of four possible locations).

Directing spatial attention in both studies led to even greater P3 responses when the stimuli changed to the attended location (see Figs 3 and 5 in this study, and see Fig 1 in [7]). At -30˚ to the left, the largest P3 response was observed for the Attend Left condition, and at +30˚ to the right, the largest peak was observed for the Attend Right condition. At non-target locations (-60˚, 0˚, and +60˚), no meaningful differences in amplitudes were observed between the two attention conditions. Variations on this paradigm have previously been used in the visual domain and have found similar results [50–52]. In the present study, cortical responses also appeared higher for active attention conditions in quiet than in babble backgrounds. This is consistent with the view that sensory gain control is dependent on signal-to-noise-ratio [53], in which adding gain to the system is predicated on the saliency of the target stimulus. Together, the results demonstrate that overall neural activity is modulated by the occurrence of a speech stream at an attended location in both quiet and babble backgrounds, but stimulus noise can interfere with the salience of location changes in target speech.

Finally, there was an observed difference between attention conditions in quiet that was not necessarily expected. In Fig 7, spatial shifts to the right led to greater evoked responses during the Attend Right condition relative to the Attend Left condition, yet shifts to the left were no different between conditions. Because the current study was not designed to investigate this effect and source analyses were not conducted to determine any cortical asymmetries, we can only speculate on this interesting observation. It may be that contralateral bias (e.g., [19]) led to an enhanced response for a rightward spatial shift in the contralateral (left) hemisphere which has also been shown to play an outsized role in speech perception [54]. In addition, endogenous spatial attention has been shown to have asymmetric hemispheric effects for auditory targets [55], consistent with the larger responses in the Attend Right condition than the Attend Left condition. It is unclear why the presence of babble would negate these effects other than the previous arguments that the salience of the location shifts was sufficiently masked to remove any hemifield effects. Future research would need to determine the relative contributions from each cortical hemifield to determine the ultimate cause of this effect.

## Conclusions

This EEG study aimed to measure the effect of active attention to a spatial location of speech in the free field, either in quiet or babble background. Earlier work demonstrated that for narrowband noise stimuli, younger and even older listeners show significant modulatory effects of attention depending on the magnitude and direction of a spatial change [7]; however, it was unknown if attention to more real-world stimuli, like speech, would show comparable modulation to evoked responses. Results demonstrated a late-latency P3 indicator of engaged attention to changes in the environment that was like the neural responses established for noise bursts. Active attention to specific target locations modulated the overall responses, and background babble at +6 dB SNR diminished responses somewhat. Still, the babble background did not eliminate the overall effect in these younger, normal-hearing listeners.

Young normal-hearing listeners are not known to have difficulty localizing speech with competing background signals at positive SNRs, but there are known challenges in individuals with hearing loss [56]. By understanding the consequences of attention on auditory evoked neural measures, it is possible that objective tasks can be designed that directly assess hearing-impaired listeners' perceptual limitations or their success with potential interventions. It is unclear, for example, whether a better SNR resulting from directional processing in hearing aids could help mediate spatial hearing challenges at the neural level. Future work will focus on the consequences of aging and hearing impairment on object-based auditory attention while evaluating the efficacy of spatial hearing enhancement (i.e., directional microphones) in hearing aids.

## Supporting information

**S1 Data.**
(XLSX)

## Acknowledgments

The authors would like to thank Abigail Biberdorf and the staff of the USF Auditory & Speech Sciences Laboratory for assistance with subject recruitment and data collection. A portion of these data were presented at the 177th Meeting of the Acoustical Society of America in Louisville, KY, and were published as proceedings of that meeting [57].

## Author Contributions

**Conceptualization:** Erol J. Ozmeral.

**Data curation:** Erol J. Ozmeral.

**Formal analysis:** Erol J. Ozmeral, Katherine N. Menon.

**Funding acquisition:** Erol J. Ozmeral.

**Investigation:** Erol J. Ozmeral.

**Methodology:** Erol J. Ozmeral.

**Project administration:** Erol J. Ozmeral, Katherine N. Menon.

**Resources:** Erol J. Ozmeral.

**Software:** Erol J. Ozmeral.

**Supervision:** Erol J. Ozmeral.

**Validation:** Erol J. Ozmeral, Katherine N. Menon.

**Visualization:** Erol J. Ozmeral.

**Writing – original draft:** Erol J. Ozmeral.

**Writing – review & editing:** Erol J. Ozmeral, Katherine N. Menon.

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
