## [Decision Letter · Decision Letter 0]

22 Jul 2022

PONE-D-22-13592Selective auditory attention modulates cortical responses to sound location change for speech in quiet and in babblePLOS ONE

Dear Dr. Ozmeral,

Thank you for submitting your manuscript to PLOS ONE. After careful consideration, we feel that it has merit but does not fully meet PLOS ONE’s publication criteria as it currently stands. Therefore, we invite you to submit a revised version of the manuscript that addresses the points raised during the review process.

In particular please address the issue of several settings not fully justified (e.g. time window/temporal location, number of subjects - with ideally a brief power analysis, choice of location of interest etc.) as raised by the reviewers. Since the work is quite experimental these are design choices that may have been made on the basis of pertinent information that would enhance the manuscript. If you choose, as I hope, to submit a revised version, I would ask that you identify new or edited sections using a different font colour. This will enable editors and reviewers to complete the review process with convenience.

We look forward to receiving your revised manuscript.

Kind regards,

Ian McLoughlin, PhD

Academic Editor

PLOS ONE

Journal Requirements:

“This work was supported by an internal grant from the College of Behavioral and Community 319 Sciences at the University of South Florida and by the NIH NIDCD to E.J.O. (R21 DC017832). ”

“This work was supported by an internal grant from the College of Behavioral and Community Sciences at the University of South Florida and by the National Institute of Deafness and Other Communication Disorders (NIDCD; https://www.nidcd.nih.gov/) to E.J.O. (R21 DC017832). The funders had no role in study design, data collection and analysis, decision to publish, or preparation of the manuscript.”

Reviewers' comments:

Reviewer's Responses to Questions

**Comments to the Author**

1. Is the manuscript technically sound, and do the data support the conclusions?

Reviewer #1: Partly

Reviewer #2: Yes

2. Has the statistical analysis been performed appropriately and rigorously? 

Reviewer #1: No

Reviewer #2: Yes

3. Have the authors made all data underlying the findings in their manuscript fully available?

Reviewer #1: Yes

Reviewer #2: Yes

4. Is the manuscript presented in an intelligible fashion and written in standard English?

Reviewer #1: Yes

Reviewer #2: Yes

5. Review Comments to the Author

Reviewer #1: Review of “Selective auditory attention modulates cortical responses to sound location change for speech in quiet and in babble”

Overall, I found the manuscript to be intriguing and detailed. However, the “exploratory” nature of the study needs to either be explicitly identified or the following changes made:

1. Description of the ERPs of interest and why/how they were elicited? In addition, what do they mean in relation to auditory attention and object formation?

2. Primary hypotheses and statistical analyses stated. The planned analyses are not stated in any section leading the manuscript to have too much of an “exploratory” feel for this journal.

Abstract:

The abstract was concise but more simplified results are needed. For example, I did not understand the sentence, “However, the effect of the two background environments diverges when considering the magnitude and direction of the location change, in which there was a clear influence of change vector in quiet but not in babble.” What was the direction in which subject performed better? Or was the subject able to change location more easily in quiet? The authors may also want to be more descriptive of cortical object formation (line 29) and what it means in respect to the cocktail party effect.

Introduction:

The premise of manuscript is setup nicely in the first three paragraphs. Previous literature supports this current work and show a need for better understanding how auditory object formation is affected by the interaction of spatial location and attention. In addition, the authors highlight the use of speech as evoking stimuli and background noise to make this experiment even more ecologically valid. My only critique of the introduction is the wording of the last paragraph. The goals and hypotheses need to be clearly outlined. Keep it simple. Also, what were the main independent variables? From my review it seems like there are three variables of interest: location, attention, and background noise.

Methods:

The procedural and electrophysiologic methodology is well defined and clear. It is obvious that an electrophysiology researcher is the primary author as someone could easily read this and replicate the study. I would have liked to see more details regarding the specific measures of the components (N1-P3). For example, were amplitude, latency, or even area measures considered? However, there is one large problem with the babble masker, they in eight foreign languages? This needs to be discussed as foreign language maskers have been shown to be basically energetic maskers especially if the talkers exceed 6. More discussion of why the authors chose foreign language maskers is warranted. The final and major critique of the methods section is the lack of statistical and power analysis section. What were the planned comparisons? This goes back to the introduction as there does not seem to be a clear analysis strategy. How were the conditions compared and what statistics were conducted? Was a linear-mixed effect model used to address the repeated measures? How many conditions and was there a correction factor for number of analyses? Also, is the study adequately powered for only 18 subjects? A brief power analysis needs to be documented to show that results are interpretable given the low n.

Results:

The results show some interesting findings, but again it seems like these analyses were exploratory in nature. For example, why was the epoch window set to 2.0s? If the N1 and P3 were components of interest why show the waveform to 2.0s? It seems like the major findings are the difference between active and passive (Active = larger amplitude), quiet and babble (quiet = larger amplitude), and the larger magnitude for the rightward direction of tart identification. Considering the first two results should have been hypothesized, the rightward direction is fascinating. A few more follow-up questions:

1. Why were only correct trials counted? What if the object was not formed? This is valuable information that could provide more insight in the babble condition especially.

2. The authors mention the P3 component in the passive condition? This is a fundamental flaw and needs to be reworded as the P3 is only elicited in an active condition. A more detailed description of the primary components of interest is warranted in the methods.

3. Minor - Paired sample t-test alpha 0.5 (line 195)? I believe this should be 0.05.

Discussion:

The discussion identified the major findings and provided a rationale for results. There were a few statements that need further justification. The authors state that, “The N1 and P2 were not modulated by attention (Line 256)? Were analyses conducted between the passive and active amplitudes of the N1 and P2? Attention does affect P1-N1-P2 responses. Also, what latency analysis did you find (Line 259) especially between classic sensory potentials to onsets? Comparing ERPs with drastically different evoking paradigms is not entirely fair. Finally, the author state, “P3 is an automatic process (Line 280)”. I completely disagree with this statement and very much recommend that the authors read the citations from Polich. As stated above, a more thorough dive into the component morphology is warranted for this manuscript to relate the results back to electrophysiological responses.

Reviewer #2: Review of Ozmeral and Palandrani, “Selective auditory attention…”

The manuscript describes a study of cortical EEG potentials evoked by changes in the location of a speech stream. The factors under study include the influence of background sound (quiet vs speech babble), the magnitude of spatial change, the attentional state of the participant (attend left, right, or to irrelevant visual), and the alignment of spatial change with attended (target) location, when applicable. The results demonstrate strong modulation of potentials by attention and by magnitude of spatial change. Effects of background sound were muted but could be extracted in a detailed analysis of amplitude for particular pairs of pre/post switch locations.

In general the manuscript is well written. The study appears clearly organized and correctly executed. Results are analyzed in a (mostly) straightforward manner that is justified, and the conclusions are reasonable.

I have no significant concerns about publication of this manuscript, although there are elements which are not very clear in the current form. These should be addressed or omitted.

General comments

My main concerns are that the analyses depicted in Fig 6 and Fig 7 are not as clear as they could be. Why focus on the time window around 550 ms?

For Fig 6: Why was it necessary to use a heat map for the pre/post location data in Fig 6? Does Fig 6 demonstrate anything interesting other than greater response for targets (i.e. post-switch location matches attended location)? The color scale of Fig 6 makes the babble results appear qualitatively different from the quiet, but comparing Figs 3 and 5 it looks like the results are similar, just modestly reduced amplitude in babble.

For Fig 7: Would like to see an independent assessment of “target” vs non-target spatial change in this analysis. Also the text’s description of the x-axis value is not clear. (I think it’s a simple enough idea, just not explained clearly; see specific comments below) Perhaps it would be useful to define a new term like “spatial change” or “signed shift magnitude” and define it quantitatively in the text? Finally, where is the discussion of the apparent asymmetry in the data (larger responses for rightward shifts in attend right but not leftward shifts during attend left? )

Specific:

Line 43: what is meant by “shunted”?

Line 183: it would be useful to know the effect of the button press itself, or of the “respond to a target” event. For example, a difference between passive and “attend” conditions is that only the latter have targets, so successful detection can only occur in those conditions. Thus, the evoked potential differences may not have anything to do with attention, per se, (or spatial attention in particular) but could mainly reflect the occurrence of “target detection” / “yes button” events.

Line 185: explain how the presence of P3 indicates stimulus factors rather than response. I suspect there is an model-based assumption hiding in this reasoning.

Line 228: may be clearer to say larger responses for post-switch locations matching target (describing the experiment) rather than “column” (describing the figure)

Line 232: This analysis is not very clearly described. I think it is as follows: all location-change events were collected and binned according to the absolute displacement (size and direction) of the change, without regard to the pre or post switch position. The average GFP amplitude is plotted against displacement in Fig 7. However, it should be noted that the analysis includes both target and non-target trials so it is not an independent assessment of the effects of stimulus location change.

Line 482: I feel there is a lot missing from the caption for fig 7. What is a “spatial-change vector?” What do the different colored lines represent? Do the plots include all changes including both “yes” and “no” trials?

Fig 7: Some values of change are more likely to include target locations than others, although none are exclusively targets. How would this result change if only non-targets were plotted?

6. PLOS authors have the option to publish the peer review history of their article (what does this mean?). If published, this will include your full peer review and any attached files.

Reviewer #1: No

Reviewer #2: No

---

## [Author Response · Author response to Decision Letter 0]

19 Sep 2022

Thank you for your feedback. Please see the attached file for item-by-item response.

---

## [Decision Letter · Decision Letter 1]

4 Jan 2023

Selective auditory attention modulates cortical responses to sound location change for speech in quiet and in babble

PONE-D-22-13592R1

Dear Dr. Ozmeral,

We’re pleased to inform you that your manuscript has been judged scientifically suitable for publication and will be formally accepted for publication once it meets all outstanding technical requirements.

Kind regards,

Xiong Jiang

Academic Editor

PLOS ONE

Additional Editor Comments (optional):

Reviewers' comments:

Reviewer's Responses to Questions

**Comments to the Author**

1. If the authors have adequately addressed your comments raised in a previous round of review and you feel that this manuscript is now acceptable for publication, you may indicate that here to bypass the “Comments to the Author” section, enter your conflict of interest statement in the “Confidential to Editor” section, and submit your "Accept" recommendation.

Reviewer #1: All comments have been addressed

2. Is the manuscript technically sound, and do the data support the conclusions?

Reviewer #1: Yes

3. Has the statistical analysis been performed appropriately and rigorously? 

Reviewer #1: Yes

4. Have the authors made all data underlying the findings in their manuscript fully available?

Reviewer #1: Yes

5. Is the manuscript presented in an intelligible fashion and written in standard English?

Reviewer #1: Yes

6. Review Comments to the Author

Reviewer #1: The responses each reviewer were incredibly detailed and changes made to the manuscript accurately reflected said changes. I was impressed by the changes especially for the addition in the methods. I believe the clarity of the manuscript greatly increased with this revision. I do not have any further comments or critiques.

Honestly, I think this is fascinating work that needs to be published and PLOS ONE is a great avenue to do that.

7. PLOS authors have the option to publish the peer review history of their article (what does this mean?). If published, this will include your full peer review and any attached files.

Reviewer #1: **Yes: **Christopher Edward Niemczak

---

## [Editor Report · Acceptance letter]

5 Jan 2023

PONE-D-22-13592R1 

Selective auditory attention modulates cortical responses to sound location change for speech in quiet and in babble 

Dear Dr. Ozmeral:

I'm pleased to inform you that your manuscript has been deemed suitable for publication in PLOS ONE. Congratulations! Your manuscript is now with our production department. 

Kind regards, 

on behalf of

Dr. Xiong Jiang 

Academic Editor

PLOS ONE